# Single-cell transcriptomes of the human skin reveal age-related loss of fibroblast priming

Llorenç Solé-Boldo [1], Günter Raddatz[1], Sabrina Schütz[1], Jan-Philipp Mallm[2], Karsten Rippe [2], Anke S. Lonsdorf[3], Manuel Rodríguez-Paredes [1,4✉] & Frank Lyko[1,4✉]

Fibroblasts are an essential cell population for human skin architecture and function. While fibroblast heterogeneity is well established, this phenomenon has not been analyzed systematically yet. We have used single-cell RNA sequencing to analyze the transcriptomes of more than 5,000 fibroblasts from a sun-protected area in healthy human donors. Our results define four main subpopulations that can be spatially localized and show differential secretory, mesenchymal and pro-inflammatory functional annotations. Importantly, we found that this fibroblast 'priming' becomes reduced with age. We also show that aging causes a substantial reduction in the predicted interactions between dermal fibroblasts and other skin cells, including undifferentiated keratinocytes at the dermal-epidermal junction. Our work thus provides evidence for a functional specialization of human dermal fibroblasts and identifies the partial loss of cellular identity as an important age-related change in the human dermis. These findings have important implications for understanding human skin aging and its associated phenotypes.

[1] Division of Epigenetics, DKFZ-ZMBH Alliance, German Cancer Research Center, 69120 Heidelberg, Germany. [2] Division of Chromatin Networks, German Cancer Research Center and Bioquant, 69120 Heidelberg, Germany. [3] Department of Dermatology, University Hospital, Ruprecht-Karls University of Heidelberg, 69120 Heidelberg, Germany. [4] These authors contributed equally: Manuel Rodríguez-Paredes, Frank Lyko. ✉email: m.rodriguez@dkfz.de; f.lyko@dkfz.de

T he skin is the outermost protective barrier of the organism and comprises two main layers, the epidermis and the dermis. The epidermis is a stratified squamous epithelium composed of keratinocytes (around 95%) and other minority cell types such as melanocytes, Merkel and Langerhans cells[1–4]. The dermis is a much thicker layer located beneath the epidermis and plays an instrumental role in skin architecture and function[5,6]. It consists mostly of the extracellular matrix (ECM) generated by its numerous fibroblasts, and also includes many other cell types due to the various structures it harbors, such as the vasculature, nerves, sweat glands, and lymphatic vessels[7]. A comprehensive molecular characterization of all skin cell types, together with the detailed knowledge of their functions and interactions is crucial to understand skin homeostasis in health and disease.

Much of our current knowledge about the cellular components of skin has been generated in mice using reporter constructs and lineage tracing, as well as fluorescence-activated cell sorting (FACS) on enzymatically digested tissue. Although limited by the use of predetermined markers, these methods, combined with immunohistochemistry (IHC) have characterized numerous cell (sub)types and defined their locations[8–10]. These approaches have also described key differences for the fibroblasts in the superficial papillary dermis and the underlying reticular dermis[9]. For example, while papillary fibroblasts are morphologically thin and spindle-shaped, reticular fibroblasts are squarer and more expanded[11,12]. Further differences include their rate of proliferation, contractility, production of and response to cytokines and growth factors, as well as the expression of ECM components such as collagens and proteoglycans[6]. Examples for the latter include Collagen alpha-1(IV) chain, which is more expressed by papillary fibroblasts, or Decorin and Versican, which are more expressed by papillary and reticular fibroblasts, respectively[13,14]. These observations suggest different roles for these two subtypes of fibroblasts and for the different histological layers they define. On the functional level, the papillary dermis[7] is known to be essential for epidermal organization due to its close interactions with keratinocytes[6,15,16]. Interestingly, it has been suggested that intrinsic skin aging may have a stronger impact on the papillary fibroblasts, eventually leading to their loss or impaired functionality[11,17] and thus contributing to the visible clinical signs of aged skin, such as reduced turgor and increased wrinkling[18].

Besides their fundamental role in skin architecture, dermal fibroblasts actively participate in cutaneous immune responses, wound healing and communication with the nervous and vascular systems[19–21]. This functional diversity is now beginning to be unveiled by single-cell RNA sequencing (scRNA-seq), which allows simultaneous profiling of the transcriptomes of thousands of individual cells. A pioneering study, based on a comparably low number of single, flow-sorted fibroblasts from mouse dorsal skin provided evidence for the heterogeneity of these cells and identified a subtype involved in the fibrotic response to injuries[22]. More recently, the analysis of 300 flow-sorted fibroblasts from young and old mice, respectively, detected two fibroblast subpopulations that became less well-defined with age, and were characterized by a reduced expression of extracellular matrix genes and a gain of adipogenic traits[23]. In human skin, a scRNA-seq study of dorsal mid-forearm samples (2742 fibroblasts) found two main fibroblast subpopulations displaying different morphology and dermal distribution (*SFRP2*⁺ and *FMO1*⁺), as well as five minor subpopulations[24]. However, this study was performed with chronically-sun-exposed skin samples from a heterogeneous group of donors, and may thus have been affected by the cellular and molecular alterations of chronic ultraviolet (UV) radiation on the skin (photoaging), which can confound data analysis[25,26]. Finally, scRNA-seq was also used to complement a transcriptomic study of bulk flow-sorted and microdissected papillary and reticular fibroblasts from mouse and human dermis[27]. Based on 184 flow-sorted cells from the abdominal skin of a single 64 year-old female donor, the analysis further suggested the existence of several fibroblast subgroups[27].

We have now analyzed more than 15,000 cells from skin samples that were obtained from a defined, sun-protected area from two "young" (25 and 27 y/o) and three "old" (53–70 y/o) male Caucasian donors. Our results indicated the presence of four main fibroblast subpopulations that could be spatially localized and displayed characteristic functional annotations, consistent with a differential 'priming' of fibroblasts into functionally distinct subgroups. Subsequent comparative analysis with the expression profiles obtained from old fibroblasts revealed that all subgroups show an age-related loss of their identities. Interestingly, old fibroblast subpopulations also expressed genes encoding specific subsets of skin aging-associated secreted proteins (SAASP) and were predicted to have decreased interactions with other skin cell types. Altogether, our work provides a comprehensive and detailed analysis of human dermal fibroblasts at the single-cell level, and provides insight into their age-related changes.

## Results

**scRNA-seq analysis of sun-protected human skin.** The anatomy of the skin can vary considerably depending on a number of endogenous and environmental factors[28,29]. In addition to the dermal changes that occur as a result of intrinsic, chronological aging, there are the effects of photoaging caused by chronic exposure of the skin to low, non-extreme doses of UV radiation, the most important cause of extrinsic skin aging[25,26,30]. To minimize confounding effects of photoaging, our scRNA-seq analysis was based on five independent whole-skin samples that were obtained specifically from the sun-protected inguinoiliac region of male donors. Since we also sought to address the effects of intrinsic aging on dermal fibroblasts, samples were obtained from two younger (25 and 27 y/o) and three older (53, 69, and 70 y/o) donors. After enzymatically and mechanically disrupting the tissue, dead cells were thoroughly removed and samples were subjected to scRNA-seq using the 10X Genomics platform (v2 chemistry). This commercial version of the high-throughput Drop-seq protocol[31] identifies cell populations by analyzing the expression of highly expressed genes in a high number of cells.

In an initial analysis, we obtained an overview of the diverse skin populations by integrating the cells from all five samples. Data analysis of the 15,457 cells that passed our quality controls (Supplementary Table 1, see Methods for details) resulted in a uniform manifold approximation and projection (UMAP)[32] plot displaying 17 clusters with distinct expression profiles (Fig. 1a). Importantly, all identified clusters contained cells from all donors (Supplementary Fig. 1a). Comparing known markers with the most representative expressed genes of each cluster (Fig. 1b and Supplementary Data 1) revealed the identity of the 17 cell clusters, all of which are known constituents of the human skin and represent nine main cell types (Fig. 1c and Supplementary Figs. 1b–d). Seven clusters comprised the two key cell types of the skin, keratinocytes and fibroblasts. Keratinocytes were detected in three clusters (#5, #7 and #15) and their diversity was mainly due to their degree of differentiation. While epidermal stem cells (EpSC) and other undifferentiated progenitors (#7 and #15) expressed markers such as *KRT5*, *KRT14*, *TP63*, *ITGA6*, and *ITGB1*, differentiated keratinocytes (#5) were defined by *KRT1*, *KRT10*, *SBSN*, and *KRTDAP* expression[33] (Fig. 1c and Supplementary Fig. 1c). Fibroblasts were identified by their archetypal markers *LUM*, *DCN*, *VIM*, *PDGFRA*, and *COL1A2*[27], constituted

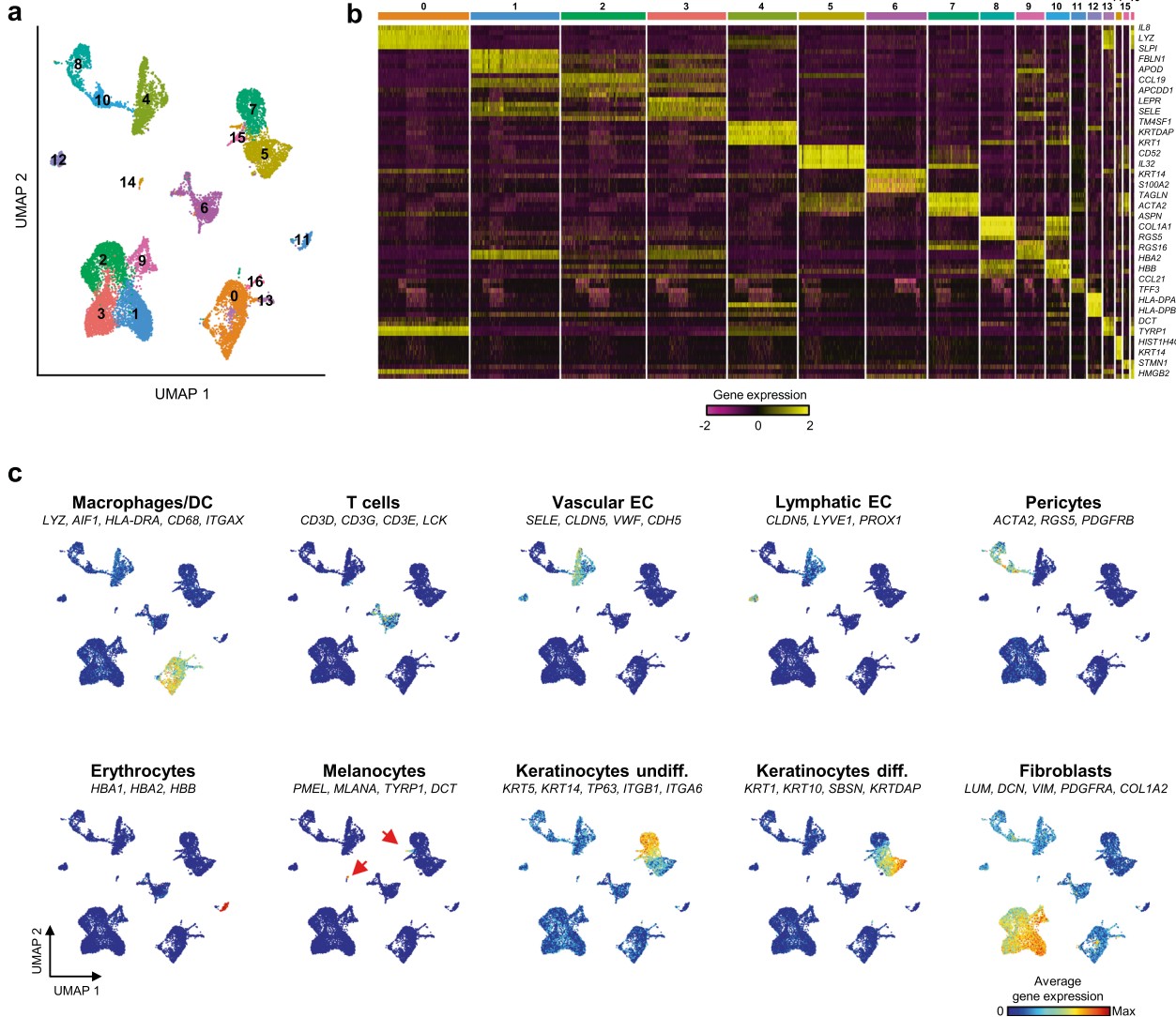

**Fig. 1 Single-cell RNA sequencing analysis of sun-protected whole human skin identifies seventeen distinct cell populations. a** Uniform manifold approximation and projection (UMAP) plot depicting single-cell transcriptomes from whole human skin ($n = 5$). Each dot represents a single cell ($n = 15,457$). Coloring is according to the unsupervised clustering performed by Seurat. **b** Heatmap showing the five most differentially expressed genes of each cell cluster, as provided by Seurat. Each column represents a single cell, each row represents an individual gene. Two marker genes per cluster are shown on the right. Yellow indicates maximum gene expression and purple indicates no expression in scaled log-normalized UMI counts. **c** Average expression of 3–5 well-established cell type markers was projected on the UMAP plot to identify all cell populations (see Methods for details). Red indicates maximum gene expression, while blue indicates low or no expression of a particular set of genes in log-normalized UMI counts. DC dendritic cells, EC endothelial cells.

the most abundant skin cell type (5948 cells in total), and were represented by four clusters (#1, #2, #3, and #9, Fig. 1c and Supplementary Fig. 1c). Despite the lower amount of cells, the individual analysis of each sample generated a similar number of clusters and identified the same major cell types (Supplementary Fig. 2).

**Functional and spatial signatures of fibroblast subpopulations**. To investigate whether specific functions could be assigned to the different fibroblast subpopulations, we performed gene ontology (GO) analyses using the most representative markers of each cluster. Since it is well established that skin and its fibroblasts undergo specific changes upon aging[34–36], we only used the expression profiles of the fibroblasts from young samples (1792 cells) for this analysis (Fig. 2a and Supplementary Data 2). Classical fibroblast functions related to collagen or ECM production and organization were strongly enriched for three of the

clusters (#1, #3, and #9, Fig. 2b). GO analyses also assigned mesenchymal functions such as skeletal system development, ossification or osteoblast differentiation to the cells belonging to clusters #3 or #9 (Fig. 2b). Interestingly, our results also showed a strong enrichment for functions related to inflammation specifically in the fibroblasts of cluster #2. Significant examples include inflammatory response, cell chemotaxis or negative regulation of cell proliferation, necessary for the final anchoring of leukocytes (Fig. 2b). Functions that are typically attributed to fibroblasts, such as collagen or ECM production and organization, did not appear among the most statistically significant categories for the cells of this cluster. These findings provide a first illustration for the functional heterogeneity of the fibroblasts in our samples.

The expression of some specific collagens has also been linked to particular fibroblast functions[37]. We therefore analyzed the four fibroblast clusters at the level of collagen expression patterns. In agreement with our GO analysis, the results clearly indicated

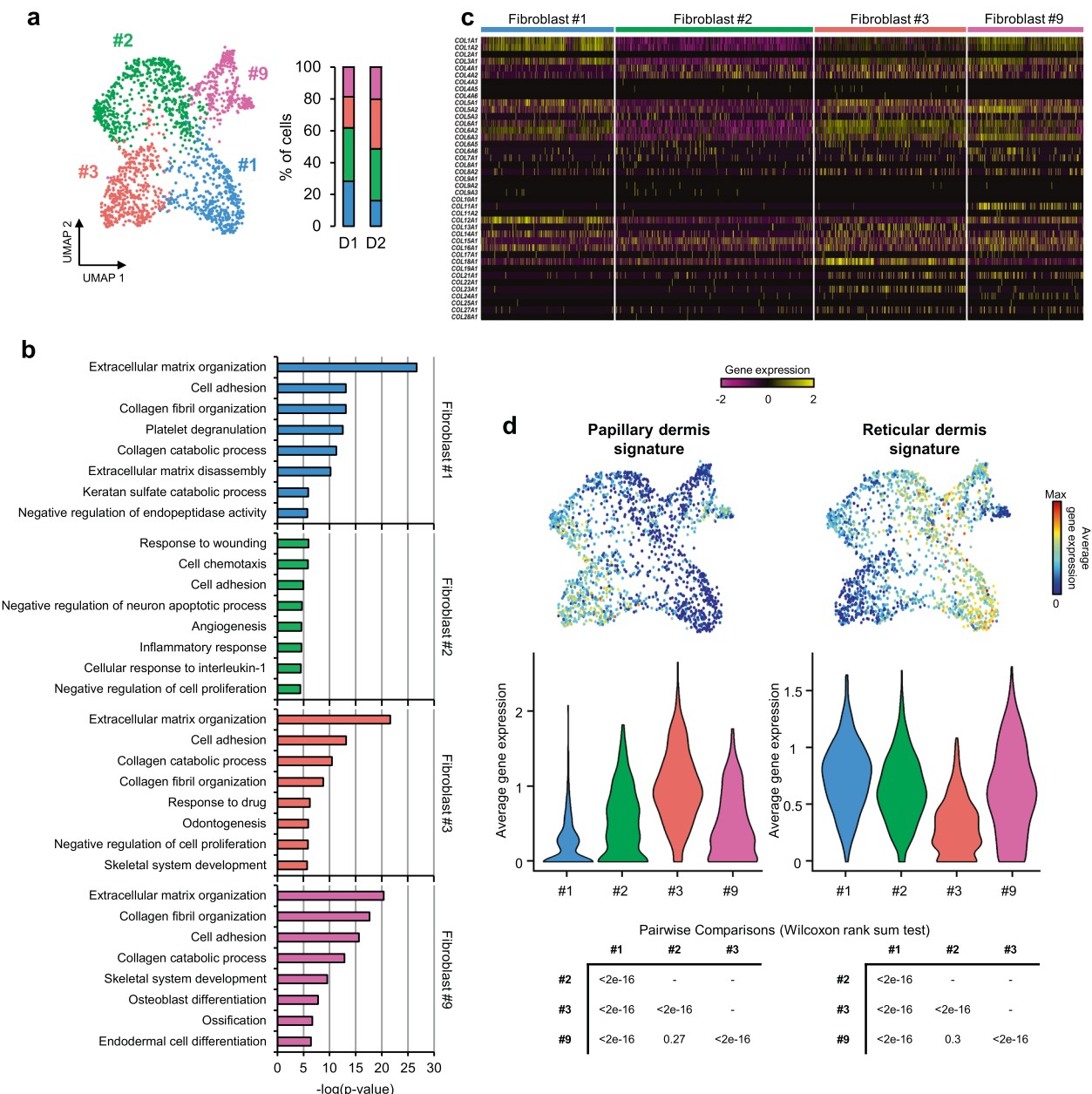

**Fig. 2 Dermal fibroblast subpopulations display specific spatial and functional transcriptomic signatures. a** Left: UMAP plot displaying dermal fibroblasts from young donors (*n* = 2). Each dot represents a single cell (*n* = 1792). Coloring is according to the original unsupervised clustering performed by Seurat. Right: Bar plots indicating the percentage of fibroblasts corresponding to each subpopulation and donor. **b** Top 8 enriched Gene Ontology (GO) terms in each fibroblast subpopulation, sorted by p-value. **c** Heatmap showing the expression of all collagen genes in the distinct fibroblast subpopulations. Each column represents a single cell and each row an individual collagen gene. Yellow indicates maximum gene expression while purple indicates no expression in scaled log-normalized UMI counts. **d** Average expression of the genes constituting the papillary and reticular gene signatures for predicting dermal localization of the fibroblasts from the four clusters. In all UMAP gene expression projections, red indicates maximum expression and blue indicates low or no expression of each particular set of genes in log-normalized UMI counts. In the violin plots, *X*-axes depict cell cluster number and *Y*-axes represent average expression of each set of genes in log-normalized UMI counts. Statistical significance of the expression changes in the gene signatures between cell clusters is indicated below through the p-values of the corresponding Wilcoxon rank sum tests.

lower global collagen expression levels in cluster #2 (Fig. 2c). The analysis also revealed that the collagen genes *COL11A1* and *COL24A1*, associated with cartilage and bone development[38,39], respectively, were specifically expressed in the fibroblasts of cluster #9 (Fig. 2c), suggesting a stronger mesenchymal component for this cell subpopulation. For the remaining collagen and ECM secreting clusters (#1 and #3), the data showed a bias in the production of collagens that have previously been linked to specific dermal locations (Fig. 2c). In particular,

fibroblasts in cluster #3 express *COL13A1* and *COL23A1*, two known markers of papillary fibroblasts[27,40–42]. High expression levels of another epidermal-dermal junction collagen gene, *COL18A1*, also supported the location of these fibroblasts within the papillary layer[27,40,43,44].

To better predict the potential localization of the four observed fibroblast subpopulations within the dermis, we next studied the expression of sets of genes that have previously been related to papillary or reticular fibroblasts. While the most representative

markers of the papillary fibroblasts comprise *APCDD1*, *AXIN2*, *COLEC12*, *PTGDS*, and *COL18A1*, the reticular fibroblast signature is typically defined by a group of ten genes, including *MGP* or *MFAP5*[11,27,40,44] (Supplementary Fig. 3). In agreement with the collagen expression data, our results showed that the papillary gene expression signature is mostly restricted to fibroblasts in cluster #3 (Fig. 2d). In contrast, fibroblasts in clusters #1, #2, and #9 showed more prominent expression of the reticular signature, with the highest reticular (and lowest papillary) expression levels observed in cluster #1 (Fig. 2d).

Taken together, these results suggest that the functional annotation differences between subpopulations reflect their priming for distinct functional roles. Our results thus predict two subpopulations with prominent roles in the generation of structural collagen and ECM organization, one located in the reticular dermis (cluster #1, 'secretory-reticular fibroblasts') and the other in the papillary dermis (cluster #3, 'secretory-papillary fibroblasts'). A third subpopulation, which was predicted to have a more reticular localization, showed a greater mesenchymal potential (cluster #9, 'mesenchymal fibroblasts'). The fourth and final subpopulation, also predicted to have a mostly reticular localization, was characterized by pro-inflammatory functions (cluster #2, 'pro-inflammatory fibroblasts').

The subdivision of fibroblasts into four subpopulations was also confirmed by a second-level clustering of the 1792 fibroblasts obtained from the young samples. This approach also identified both secretory and mesenchymal subpopulations, while further subdividing the pro-inflammatory subpopulation into two closely related subclusters that were defined by the differential expression of a subset of genes (Supplementary Fig. 4a and Supplementary Data 3). Second-level analyses of the fibroblasts from each individual sample (young or old) also often subdivided the pro-inflammatory subpopulation and additionally separated the well-established dermal papilla-associated fibroblasts, which are characterized by high *CRABP1* and *TNN* expression levels[9,24], from the mesenchymal subpopulation (Supplementary Figs. 4b, c). However, these additional subclusters are clearly related to the four main fibroblast subpopulations and were therefore not considered separately.

**Validation of fibroblast subpopulations in skin sections**. To further characterize and validate the four fibroblast subpopulations from our initial analysis, we identified the most representative markers for each subpopulation according to their expression in the specific cell clusters (Table 1 and Supplementary Fig. 4a). Since no cell-surface markers were found specific enough for all subpopulations, to assess the microanatomical distribution of the characterized subpopulations we then performed RNA FISH on independent, formalin-fixed paraffin-embedded (FFPE) skin sections from young (28–37 y/o) and old (54-86 y/o) donors. *Collagen Triple Helix Repeat-Containing 1* (*CTHRC1*), and *Adenomatosis polyposis coli downregulated 1* (*APCDD1*) were selected to detect the secretory-reticular and secretory-papillary fibroblasts, respectively. *C-C motif chemokine ligand 19* (*CCL19*) and *Apolipoprotein E* (*APOE*), were used to detect the pro-inflammatory fibroblasts, and *Asporin* (*ASPN*) was chosen as a marker for the mesenchymal subpopulation. *Platelet-derived growth factor receptor alpha* (*PDGFRA*) was included as a pan-fibroblast control[45]. RNA FISH experiments confirmed the location of each secretory subpopulation within the papillary and the reticular dermal layers, respectively (Fig. 3a and Supplementary Fig. 5a). The locations were also confirmed by immunofluorescence staining of Tetraspanin 8 and the Collagen alpha-1(XVIII) chain, two additional markers of the secretory subpopulations (Table 1, Supplementary Figs. 4a and 6a, b). The pro-inflammatory fibroblasts showed a more widespread distribution and a preferential association with the

**Table 1 Representative marker genes of each fibroblast subpopulation.**

|  | Gene | Fold-change | % cells in cluster | % cells not in cluster | Adjusted *p*-value |
|---|---|---|---|---|---|
| Secretory-reticular | WISP2 | 15.06 | 0.83 | 0.074 | 0 |
|  | SLPI | 11.64 | 0.729 | 0.065 | 0 |
|  | CTHRC1 | 7.69 | 0.759 | 0.092 | 0 |
|  | MFAP5 | 0.84 | 0.474 | 0.057 | 1.29E-259 |
|  | TSPAN8 | 4.28 | 0.569 | 0.056 | 3.44E-107 |
| Pro-inflammatory | CCL19 | 12.51 | 0.343 | 0.096 | 3.79E-75 |
|  | APOE | 8.484.70 | 0.868 | 0.281 | 3.59E-275 |
|  | CXCL2 | 4.61 | 0.698 | 0.39 | 2.13E-80 |
|  | CXCL3 | 4.35 | 0.525 | 0.238 | 3.77E-63 |
|  | EFEMP1 | 3.12 | 0.564 | 0.126 | 6.36E-167 |
| Secretory-papillary | APCDD1 | 6,03 | 0.78 | 0.11 | 0 |
|  | ID1 | 3.81 | 0.60449 | 0.187 | 1.80E-109 |
|  | WIF1 | 3.74 | 0.438 | 0.035 | 3.01E-232 |
|  | COL18A1 | 2.96 | 0.581 | 0.168 | 1.68E-113 |
|  | PTGDS | 2.94 | 0.559 | 0.196 | 2.05E-152 |
| Mesenchymal | ASPN | 8.75 | 0.666 | 0.067 | 7.31E-291 |
|  | POSTN | 5.44 | 0.620 | 0.104 | 2.46E-170 |
|  | GPC3 | 3.58 | 0.513 | 0.063 | 2.83E-177 |
|  | TNN | 3.42 | 0.337 | 0.007 | 2.10E-286 |
|  | SFRP1 | 3.26 | 0.406 | 0.040 | 5.61E-165 |

The table shows the five genes selected as marker genes for each fibroblast subpopulation according to their fold-change and enriched expression in comparison with the other subpopulations.

vasculature (Fig. 3b and Supplementary Fig. 5b). Finally, the mesenchymal subpopulation was localized mostly to the reticular dermis, particularly in the vicinity of hair follicles (Fig. 3c and Supplementary Fig. 5c). This localization was also confirmed by immunofluorescence staining of Periostin, which is another marker for this subpopulation (Table 1, Supplementary Figs. 4a and 6c). Together, these results provide important confirmation for our findings obtained by single-cell transcriptomics and establish markers for the detection of specific fibroblast subpopulations.

**Aging leads to loss of dermal fibroblast priming**. We also investigated the effect(s) of aging at the level of dermal fibroblast subpopulations. In relative terms, our results suggest an apparent reduction in the number of mesenchymal fibroblasts in the old samples (Figs. 2a and 4a). However, this observation could not be experimentally demonstrated due to the low amount of hair follicles present in the available tissue sections. Next, we compared the fibroblast transcriptomes from old donors with their young counterparts. In agreement with the reduced proliferative capacity of aged cells, the expression profiles from old fibroblasts indicate a significant delay at the G1/S transition of the cell cycle in the pro-inflammatory and secretory-papillary subpopulations, as well as a similar tendency for the secretory-reticular cells (Fig. 4b). Interestingly, GO analyses of the most representative genes from the old subpopulations (Supplementary Data 4) suggested a considerable age-dependent loss of the functional annotations for each cluster. In comparison to young fibroblast subpopulations, the aged counterparts showed fewer function-related terms, and/or substantially reduced p-values, consistent with fewer genes supporting the terms (Fig. 4c). A similar effect was also seen at the level of collagen gene transcription, which became particularly decreased in the secretory clusters (Fig. 4d and Supplementary Figs. 7 and 8a). Finally, aging also changed the previously defined spatial gene expression signatures, as old papillary fibroblasts presented less papillary and more reticular gene expression signatures, while reticular fibroblasts presented a less pronounced reticular gene expression signature (Fig. 4e and Supplementary Fig. 8b). These findings are in agreement with an age-related loss of fibroblast priming.

**Aging effects on SAASP profiles and cell–cell interactions**. Finally, we also analyzed whether age-related transcriptomic

**a** Secretory-reticular and secretory- papillary

**b** Pro-inflammatory

**c** Mesenchymal

changes in fibroblast subpopulations could explain age-related skin phenotypes. For example, it is known that aged fibroblasts become more susceptible to the accumulation of reactive oxygen species[46]. Consistently, GO analyses performed with the most downregulated genes of each aged cluster (Supplementary Data 5) showed that genes related to hydrogen peroxide metabolism were decreased in three of the four fibroblast subpopulations

(Supplementary Fig. 9). Furthermore, old skin is known to acquire a chronic, low-grade inflammatory phenotype[47,48]. This could also be detected at the level of fibroblast subpopulations, as we found immune response among the main enriched terms in our GO analyses of the most up-regulated genes of each aged cluster (Supplementary Figs. 9 and 10 and Supplementary Data 5). Old fibroblasts also showed changes in the expression of

**Fig. 3 RNA FISH detection of fibroblast subpopulations in young skin. a** Representative confocal images showing mRNA expression of *CTHRC1* (green) and *APCDD1* (red), selected markers for the secretory-reticular and secretory-papillary fibroblast subpopulations, respectively. Details from the papillary and reticular regions of the images above are shown in the lower panels (left and center, respectively), and percentage of positive cells for each gene and per dermal region are shown in the lower right panel. **b** Representative confocal images showing mRNA expression of *CCL19* (green) and *APOE* (red), selected markers for the pro-inflammatory fibroblast subpopulation. A detail of a vessel of the images above is shown in the lower panel. **c** Representative confocal images showing mRNA expression of *ASPN* (green), selected marker for the mesenchymal fibroblast subpopulation. A detail of the hair follicle bulb of the images above is shown in the lower panel. Dashed lines in **a** and **b** denote the papillary dermis regions while in **c** denote the dermal papilla. Nuclei were counterstained with DAPI. Each assay was performed in three independent young FFPE skin sections (28–37 y/o). Images are shown at ×40 original magnification. Scale bar: 50 µm for main images and 10 µm for detail images. Pap papillary dermis, Ret reticular dermis, Deep ret deep reticular dermis, HF hair follicle, DP dermal papilla. Statistical analyses were performed using a two-way ANOVA test (*p < 0.05, **p < 0.01, ***p < 0.001, ****p < 0.0001); error bars represent the standard deviation.

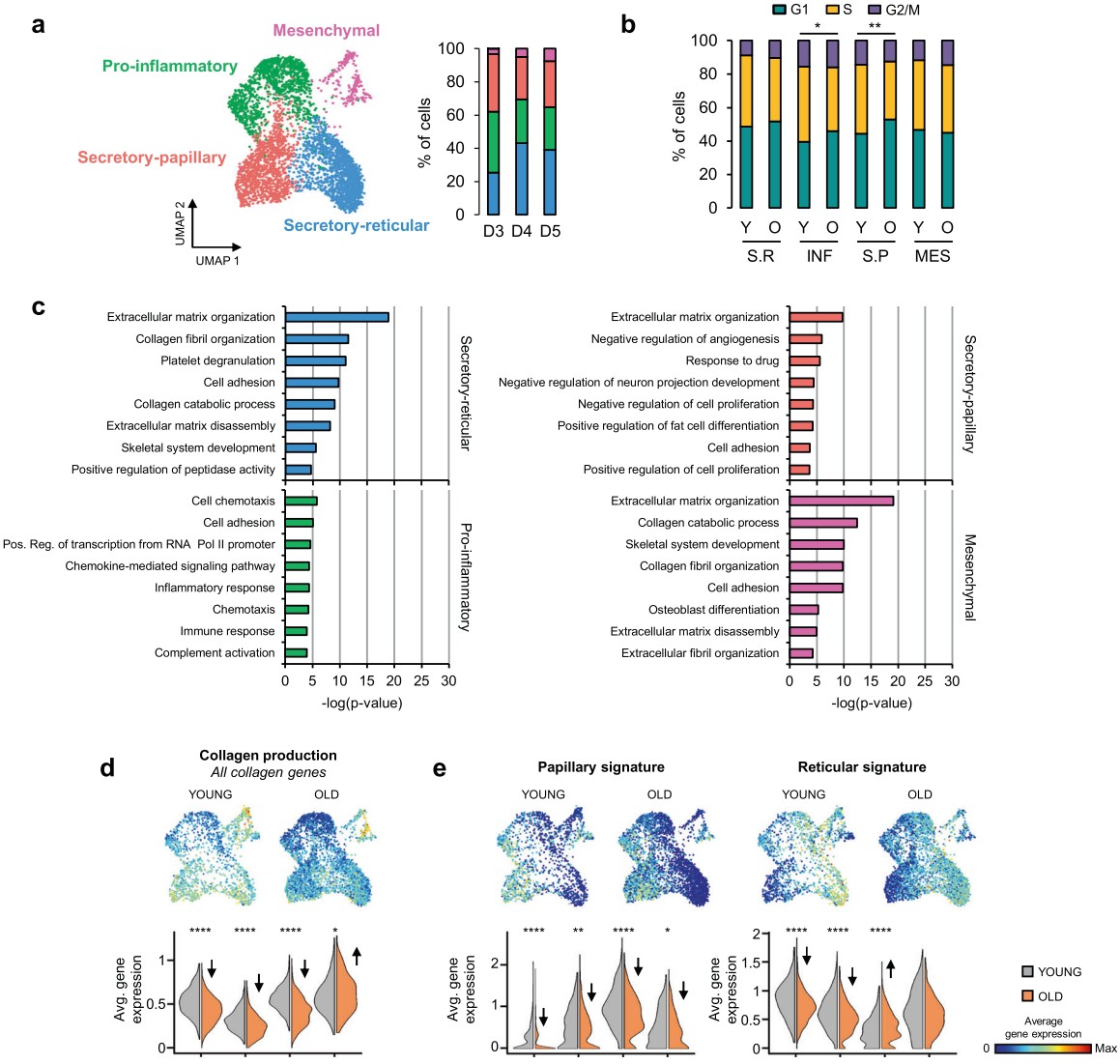

**Fig. 4 Aging leads to loss of dermal fibroblast priming. a** Left: UMAP plot displaying dermal fibroblasts from old donors (n = 3). Each dot represents a single cell (n = 4156). Coloring is according to the original unsupervised clustering performed by Seurat. Right: Bar plots indicate the percentage of fibroblasts corresponding to each subpopulation and donor. **b** Percentage of fibroblasts of each subpopulation that were in the G1, S or G2/M phase of the cell cycle in young and old skin samples, respectively. **c** Top 8 enriched Gene Ontology (GO) terms in each old fibroblast subpopulation sorted by p-value. Coloring is according to the unsupervised clustering performed by Seurat. **d** UMAP and violin plots displaying the average expression of all collagen genes in the fibroblasts of all subpopulations, for young and old skin. **e** UMAP and violin plots displaying the expression of the papillary and reticular gene signatures in the fibroblasts of all subpopulations, for young and old skin. In all UMAP gene expression projections, red indicates maximum expression and blue indicates low or no expression of each particular set of genes in log-normalized UMI counts. In the violin plots, *X*-axes depict fibroblast subpopulations and *Y*-axes represent average expression of each set of genes in log-normalized UMI counts. For comparing the ratio of G1 cells between young and old subpopulations, a two-sided two-proportion z-test was used (**b**). Statistical analyses in **d** and **e** were performed using the Wilcoxon Rank Sum test. *p < 0.05, **p < 0.01, ***p < 0.001, ****p < 0.0001. Y young, O old, S.R Secretory-reticular, INF Pro-inflammatory, S.P Secretory-papillary, MES Mesenchymal.

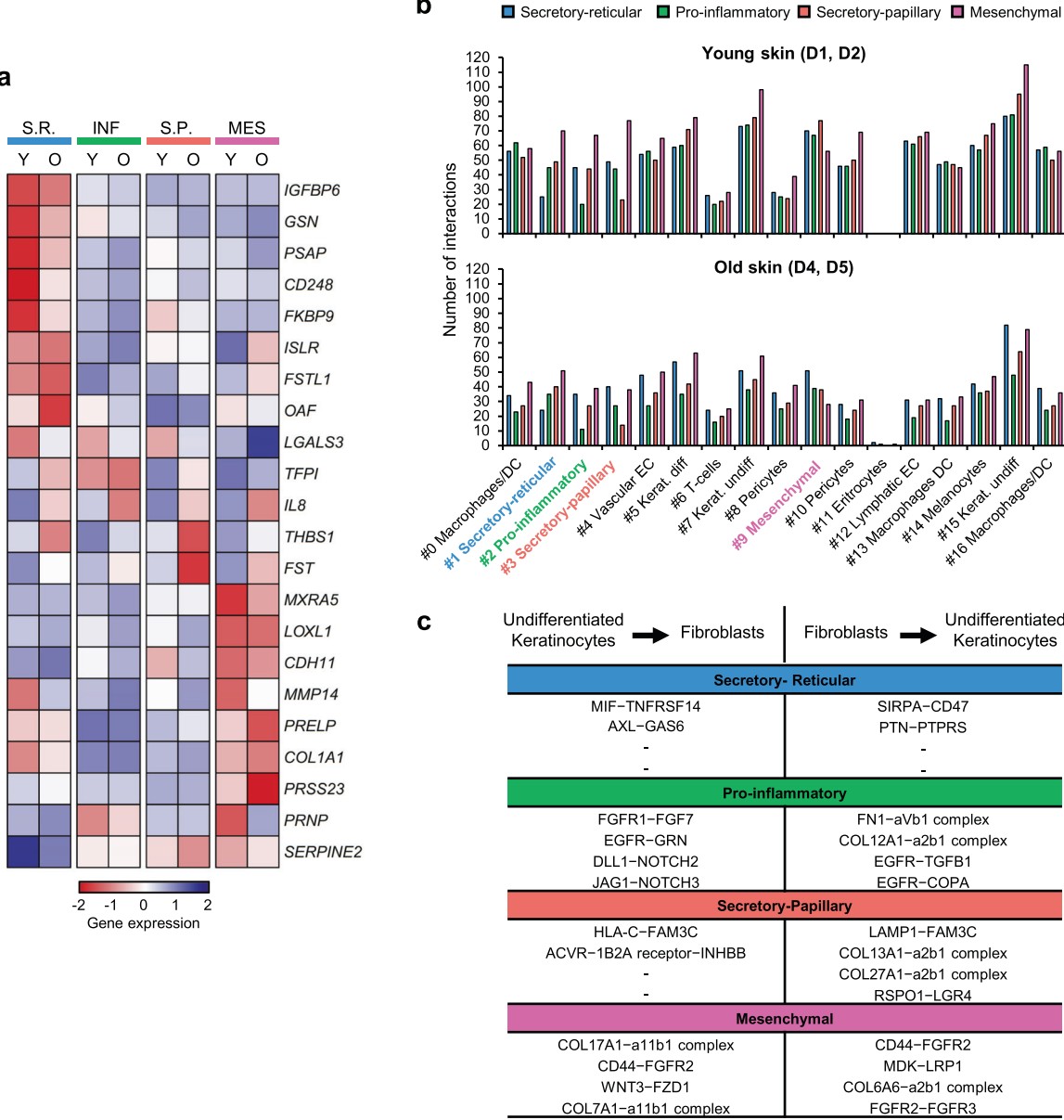

**Fig. 5 Other age-related changes in dermal fibroblast subpopulations. a** Expression of genes encoding skin aging-associated secreted proteins (SAASP) (rows) that are differentially (fold-change > 1.25) expressed between young and old fibroblasts in at least one subpopulation (columns). The heatmap shows the mean relative expression by cluster. **b** Bar plots showing the number of ligand-receptor interactions predicted for the four observed fibroblast subpopulations with the rest of the cell types identified in human skin, in the two young (≤27 y/o) (up) and the two oldest (≥69 y/o) (down) samples. The medium-aged sampled (53 y/o) showed an intermediate phenotype in this analysis and was therefore omitted. Coloring and numbering are according to the original unsupervised clustering performed by Seurat. **c** Summary of the top four exclusive interactions lost between each fibroblast subpopulation and undifferentiated keratinocytes, sorted by p-value. The table shows interactions in both directions for each pair. Y young, O old, S.R Secretory-reticular, INF Pro-inflammatory, S.P Secretory-papillary, MES Mesenchymal.

genes that were previously detected in the aging dermal fibroblast secretome[49]. Furthermore, differential expression profiles of SAASP genes could also be observed in the different old fibroblast subpopulations (Fig. 5a and Supplementary Fig. 11). In conclusion, our scRNA-seq data thus recapitulates known phenotypes associated with skin aging.

Finally, fibroblasts are known to establish interactions with many other skin cell types during homeostasis[6]. Importantly, scRNA-seq also provides novel opportunities to identify communicating pairs of cells based on the expression of cell-surface receptors and their interacting ligands[50]. Interestingly, our results indicate that a high number of the interactions predicted for

young fibroblasts are lost during intrinsic skin aging. This effect was particularly pronounced in the two oldest samples (≥69 y/o) and for interactions involving undifferentiated keratinocytes (Fig. 5b, c and Supplementary Fig. 12). These findings suggest that the loss of interactions between fibroblasts and their communicating cell types represent a previously unrecognized molecular phenotype of the aging human skin.

## Discussion

Single-cell transcriptomics currently represents the most effective method to define cell populations in a given tissue[51,52]. However,

whole human skin has not been fully analyzed yet, as previous single-cell studies were either focused on sun-exposed material obtained from a heterogeneous group of donors[24] or provided limited coverage from flow-sorted cells[27]. Our study analyzes single-cell transcriptomes from more than 15,000 skin cells, including more than 5000 fibroblasts, which were all obtained from the same sun-protected location from healthy male donors. This allowed us to minimize confounding effects and provide what we believe is the first description of intrinsic age-related changes in human dermal fibroblasts.

We describe an overall cellular composition of the human skin samples that is consistent with previous observations[24]. In addition, we also describe several fibroblast subpopulations with distinct functional annotations and spatial localizations in the dermis. These findings provide an important extension of previous observations describing fibroblast heterogeneity, including the distinction of papillary and reticular fibroblasts[6,9,45]. More specifically, our results suggest the existence of four major dermal fibroblast subpopulations: secretory-reticular, secretory-papillary, pro-inflammatory, and mesenchymal fibroblasts.

Interestingly, the functional annotations of the four subpopulations all reflect known functions of fibroblasts. For example, the secretion of collagens and ECM components is considered as the defining function of dermal fibroblasts, with well-known differences in the secretory activities of papillary and reticular fibroblasts. Consistently, our results define secretory-papillary and secretory-reticular fibroblasts as two separate subpopulations. Similarly, it is well known that fibroblasts can differentiate into other mesenchymal cell types[53,54], which is reflected by the predicted attributes of our mesenchymal subpopulation. Finally, the pro-inflammatory functions of dermal fibroblasts are also well established[55] and correspondingly reflected in the transcriptomic profiles of our pro-inflammatory fibroblast subpopulation. Our results thus suggest that human dermal fibroblast heterogeneity can be explained by the existence of subpopulations that are primed for different functions.

While our secretory-papillary and secretory-reticular fibroblasts were defined by the expression of previously established markers for these two dermal layers, the pro-inflammatory fibroblasts presented a mixed signature that was supported by their widespread localization within the dermis in validation experiments. These findings are in agreement with the notion that the entire dermis may require the protective function of pro-inflammatory fibroblasts. Similarly, the mesenchymal fibroblasts also displayed a more mixed localization signature, which may indicate different localizations of specific subpopulations. For example, dermal papilla-associated fibroblasts expressed a papillary dermis signature (Supplementary Fig. 13), consistent with their shared origin with papillary fibroblasts[9].

We also notice some differences to previous studies. For example, Tabib et al. described only two major fibroblast subpopulations that were defined by the expression of *SFRP2* and *FMO1*, as well as five additional, minor, closely related subpopulations[24]. However, these subpopulations were not functionally defined and their significance remained unclear. In our dataset, expression of *FMO1* was very low in all dermal fibroblasts, while *SFRP2* was expressed by both secretory subpopulations and a subgroup of the pro-inflammatory subpopulation (Supplementary Fig. 14a). While the reasons for these discrepancies remain to be elucidated, it is possible that the lower number of cells, in combination with sampling from a different, sun-exposed region (dorsal forearm) may have resulted in a less accurate stratification of fibroblast subpopulations. Furthermore, the scRNA-seq analysis performed by Philippeos et al. with 184 flow-sorted fibroblasts from a single abdominal skin sample detected five subpopulations[27]. The two major subpopulations

expressed markers that might localize them in different dermal layers, but the significance of the minor subpopulations remained unclear. While one of these subpopulations comprised only five cells, two others appeared to contain pericytes and pre-adipocytes, respectively[27]. In our fibroblasts most of the genes that were used to define those five subpopulations did not show significant expression levels, which may be attributed again to the fundamental differences existing between both experimental approaches (Supplementary Fig. 14b).

Our results also suggest an age-related loss of fibroblast priming. This was detectable both at the level of genes defining their functional annotations, and in the expression of their spatial localization signatures. These findings provide an important complement to a recent study that described age-related identity loss in murine fibroblasts[23]. While we also observed an upregulation of genes related to immune response and inflammation, we did not detect an upregulation of adipogenesis genes (Supplementary Fig. 15). These similarities and differences are likely explained by the limited evolutionary conservation of mouse and human skin[56].

Finally, fibroblasts maintain various paracrine interactions with other skin cell types, as well as direct cell–cell interactions[6,45]. For instance, their contacts along the dermal-epidermal junction with the epidermal stem and progenitor cells (EpSPCs) are key for proper epidermal homeostasis[6,15,16]. Importantly, our analysis of the interactome of each fibroblast subpopulation indicates that aging causes a considerable decrease in their potential interactions, including interactions with undifferentiated keratinocytes. This may represent a previously unknown molecular feature of skin aging. Taken together, our study thus reveals an important pattern of fibroblast heterogeneity at the level of cellular subpopulations and provides novel insight in the role of fibroblasts in skin aging.

## Methods

**Clinical samples**. Skin specimens for single-cell RNA sequencing (see Supplementary Table 1) were obtained from patients undergoing routine surgery at the Department of Dermatology, University Hospital of Heidelberg. Only remnant, clinically healthy skin, not required for diagnostic purposes, was analyzed after written informed consent by the patient and as approved by the Ethics Committee of Heidelberg University (S-091/2011) in compliance with the current legislation and institutional guidelines. All patients underwent a full body skin examination by a dermatologist prior to surgery and medical records were reviewed with a particular focus on skin diseases and/or skin-affecting co-morbidities. No clinical evidence or a history of an inflammatory or systemic skin disease (e.g., systemic sclerosis, lupus erythematosus), co-morbidities typically affecting the skin and/or UV-sensitivity of the skin (e.g., chronic immunosuppression, chronic renal failure) was recorded. Furthermore, no patient had a history of UV-therapy, showed clinical evidence of acute or chronic actinic skin damage or presented tanned skin in the inguinoiliac region at the time of surgery.

An independent set of skin samples was used for validation experiments. Old skin specimens for mRNA fluorescence in situ hybridization and immunohistochemistry were also obtained from patients undergoing routine surgery at the Department of Dermatology of Heidelberg University Hospital, and belonged to both sun-exposed and non-sun-exposed body areas. Young skin specimens were purchased from Genoskin (France) as FFPE sections or again obtained from the Department of Dermatology, University Hospital of Heidelberg. These samples were taken from non-sun-exposed body areas of healthy (male and female) individuals with no present co-morbidities.

**Single-cell RNA sequencing**. For each experiment, 4-mm punch biopsies were obtained from healthy whole-skin specimens, immediately after resection from the inguinoiliac region of five male subjects. Donors' characteristics are summarized in Supplementary Table 1. Samples were kept in MACS Tissue Storage Solution (Miltenyi Biotec, cat. no. 130-100-008) for no longer than 1 h before their enzymatical and mechanical dissociation with the Whole Skin Dissociation kit for human material (Miltenyi Biotec, cat. no. 130-101-540) and the Gentle MACS dissociator (Miltenyi Biotec), following the manufacturer's instructions. Cell suspensions were then filtered through 70-µm cell strainers (Falcon) and depleted of apoptotic and dead cells with the Dead Cell Removal Kit (Miltenyi Biotec, cat. no. 130-090-101).

Sequencing libraries were subsequently prepared following the Drop-seq methodology[31], using a Chromium Single Cell Controller and the v2 chemistry from 10X Genomics (cat. no. 120237). Thus, ~20,000 cells per sample were mixed with the retrotranscription reagents and pipetted into a Chip A Single Cell (10X Genomics, cat. no. 1000009), also containing the Single Cell 3' Gel Bead suspension and Partitioning Oil. The Chip was subsequently loaded into a Chromium Single Cell Controller (10X Genomics) where the cells were captured in nanoscale droplets containing both the reagents needed for reverse transcription and a gel bead. Resulting gel bead-in-emulsions (GEMs) were then transferred to a thermocycler in order to perform the retrotranscription, following the manufacturer's protocol. Each gel bead contained a specific 10X Genomics barcode, an Illumina R1 sequence, a Unique Molecular Identifier (UMI) and a poly-dT primer sequence. Therefore, from poly-adenylated mRNA the reaction produced full-length cDNA with a unique barcode per cell and transcript, which allowed tracing back all cDNA coming from each individual cell. Following an amplification step, cDNA was further processed by fragmentation, end repair and A-tailing double-sided size selection using AMPure XP beads (Beckman Coulter, cat. no. A63881). Finally, Illumina adaptors and a sample index (10X Genomics, cat. no. 120262) were added through PCR using a total number of cycles adjusted to the cDNA concentration. After sample indexing, libraries were again subjected to double-sided size selection. Quantification of the libraries was carried out using the Qubit dsDNA HS Assay Kit (Life Technologies), and cDNA integrity was assessed using D1000 ScreenTapes (Agilent Technologies). Paired-end (26 + 74 bp) sequencing (100 cycles) was finally performed with a HiSeq 4000 device (Illumina).

**Data analysis**. Raw sequencing data was processed with Cell Ranger, version 2.1.0, from 10X Genomics. For downstream analysis of the data we used the Seurat package version 3.1.1[57] in R version 3.5.1[58]. 16,062 cells passed the quality control steps performed by Cell Ranger. To remove possible cell doublets, we filtered out cells with more than 7500 expressed genes, and to remove potential apoptotic cells we discarded cells with more than 5% mitochondrial reads. The application of these filters resulted in a final dataset of 15,457 single-cell transcriptomes.

In order to account for inter-individual differences and correct for batch effects, we combined all our samples using the standard integration protocol described in Seurat. In a first step, we performed standard pre-processing of each dataset independently. This individual pre-processing included log-normalization of the UMI counts and the identification of the 2000 more variable genes per sample. Next, we used the function FindIntegrationAnchors() with default parameters and 30 canonical correlation analysis (CCA) dimensions to identify the integration anchors between our five datasets. These anchors were subsequently used for integration using the IntegrateData() function, again with the first 30 CCA dimensions and default parameters.

The integrated data were then used for standard cell clustering and visualization with Seurat, which uses the 2000 most variable genes of the integrated dataset as input. First, data were scaled using the ScaleData() function and principal component analysis (PCA) dimensions were calculated with the RunPCA() function. Next, unsupervised clustering of the data was performed with the FindNeighbors() and FindClusters() functions. For the FindNeighbors() function, we used the first 20 PCA dimensions to construct a Shared Nearest Neighbor (SNN) Graph for our dataset. Then, we clustered the cells with the function FindClusters() using a shared nearest neighbor (SNN) modularity optimization-based clustering algorithm with a resolution of 0.4. Finally, for visualization, we used the RunUMAP() function with default parameters and 20 PCA dimensions.

To identify genes with enriched expression in each cell cluster we used the FindAllMarkers() function in the integrated dataset. This function uses a Wilcoxon Rank Sum test to identify the representative genes of each cluster. These representative genes were used to establish the cell identity of each cluster, together with markers found in the literature for cell types typically present in the human skin. Average expression of a particular set of marker genes was used for cell type identification and was projected into UMAP or violin plots. Gene expression signatures used for the definition of cell populations were: *ACTA2, RGS5* and *PDGFRB* (pericytes, clusters #8 and #10)[59]; *KRT5, KRT14, TP63, ITGB1*, and *ITGA6* (epidermal stem cells and other undifferentiated progenitors, clusters #7 and #15); *KRT1, KRT10, SBSN*, and *KRTDAP* (differentiated keratinocytes, cluster #5); *PDGFRA, LUM, DCN, VIM*, and *COL1A2* (fibroblasts, clusters #1, #2, #3 and #9); *AIF1, LYZ, HLA-DRA, CD68*, and *ITGAX* (macrophages and dendritic cells (DC), clusters #0, #13 and #16)[60]; *CD3D, CD3G, CD3E*, and *LCK* (T cells, cluster #6)[61]; *SELE, CLDN5, VWF*, and *CDH5* (vascular endothelial cells, cluster #4)[62]; *PROX1, CLDN5*, and *LYVE1* (lymphatic endothelial cells, cluster #12)[63]; *HBA1, HBA2*, and *HBB* (erythrocytes, cluster #11)[64] and *PMEL, MLANA, TYRP1*, and *DCT* (melanocytes, cluster #14)[65].

For the second-level clustering of the fibroblasts (Supplementary Figs. 4a, b) we subsetted the cells and ran again the functions FindNeighbors() and FindClusters(). In this case, we used 20 PCA dimensions for both functions and a resolution of 0.5 for the FindClusters() function. For visualization we re-calculated the UMAP plot with RunUMAP() function with default parameters and using 20 PCA dimensions.

Aiming to infer age-related differences, we first used the FindAllMarkers() function to identify those genes whose expression is enriched in each cell cluster of the young and old skin datasets separately. To obtain the genes differentially expressed by each fibroblast cluster upon aging we used the FindMarkers() function.

For the gene ontology (GO) analyses, representative genes expressed by each fibroblast cluster in the young and old datasets, or genes differentially expressed in each cluster upon aging, were queried into the Gene Functional Annotation Tool from the DAVID Bioinformatics Database (version 6.8). GO option GOTERM_BP_ALL was selected and the first 8 GO terms with a *p*-value < 0.05 were chosen as significant categories.

Average gene expression for collagen genes and for the sets of genes defining spatial identities was calculated for young and old fibroblast subpopulations and projected onto UMAP or violin plots. To test for significance, a Wilcoxon Rank Sum test was used.

To analyze the putative cell–cell interactions established by the distinct cell types present in the human skin we used the publicly available repository CellPhoneDB v2.0.0[50]. This approach performed pairwise comparisons between all cell clusters present in the young and old skin datasets. We also analyzed each sample individually. Only receptors or ligands expressed by at least 10% of the cells in a given cluster were used for the analysis. We used 1000 iterations for the young and old datasets and 100 iterations for the individual analyses. Interactions with a *p*-value < 0.05 were selected as significant.

**RNA Fluorescence in situ hybridization (RNA FISH)**. FFPE blocks of young (28–37 y/o) and old (54–89 y/o) skin donors, fixed in 4–10% formalin and cut into 4 μm sections, were subjected to RNA FISH using the RNAScope Multiplex Fluorescent Detection Kit v2 (ACDBio, cat. no. 323100) following manufacturer's instructions with extended pretreatment of the samples. These pretreatment steps included a 15 min incubation with hydrogen peroxide, a mild boil (98–102 °C) for 30 min with target retrieval reagents and a 30 min incubation at 40 °C in the HybEZ™ (ACDBio) oven with Protease Plus (all solutions as included in the kit). Probes against human *CTHRC1* (ACDBio, cat. no. 413331), *APCDD1* (ACDBio, cat. no. 535851-C2), *CCL19* (ACDBio, cat. no. 474361), *APOE* (ACDBio, cat. no. 433091-C2), *ASPN* (ACDBio, cat. no. 404481), *CD248* (ACDBio, cat. no. 542501), and *PDGFRA* (ACDBio, cat. no. 604481-C3) mRNA molecules were used. Nuclei were counterstained with DAPI and mounted using ProLong Gold Antifade Mountant (ThermoFisher, cat. no. P36930). We performed the experiment in three young and three old sections for each target gene, always including *PDGFRA* as a pan-fibroblast marker. Images were taken with a TCS SP5 confocal microscope (Leica Microsystems) using 40X oil immersion lens and were further processed using the Fiji software[66].

*APCDD1*- and *CTHRC1*-positive cells were quantified in two images per dermal region (papillary dermis, reticular dermis and deep reticular dermis) for each skin section in young and old samples. Statistical analysis of the quantification of *APCDD1*- and *CTHRC1*-positive cells was performed using a two-way ANOVA test with Dunnett correction, which compared the percentage of positive cells in the reticular and deep reticular areas to the percentage present in the papillary area for each gene independently.

*CD248* positive cells were quantified in two images of deep reticular dermis for each skin section in young and old samples, respectively. Statistical analysis was performed using an unpaired two-sided *t*-test, comparing the percentage of *CD248* positive cells.

**Immunohistochemistry (IHC)**. IHC assays were performed with six samples aged 37–79 y/o for Tetraspanin 8, three samples aged 51–66 y/o for Periostin, and one sample of 37 y/o for the Collagen alpha-1(XVIII) chain. In all cases, remnant, healthy whole-skin specimens were fixed overnight in 4% formalin in PBS, paraffin-embedded and cut into 4 μm sections. Then, sections were deparaffinized in xylene and rehydrated in a gradient of ethanol and distilled water prior to heat-induced antigen retrieval. To that aim, slides were incubated for 30 min at 95 °C in a water bath in 10 mM Citrate buffer (pH 6.0) containing 0.05% Tween-20.

For Tetraspanin 8 and Periostin staining, skin sections were permeabilized by incubation with 0.4% Triton-X in 1% Normal Goat Serum (NGS) for 10 min, twice. Subsequently, non-specific antibody binding was blocked by incubation with 10% NGS for 1 h, followed by overnight incubation with primary antibodies diluted in blocking solution at 4 °C. Primary antibodies used were rabbit anti-Tetraspanin 8 (Abcam, Ab230448, 1:200), mouse anti-Periostin (Santa Cruz, sc-398631, 1:100), rabbit anti-Vimentin (Cell Signaling, D21H3, 1:100) and chicken anti-Vimentin (Abcam, Ab24525, 1:2000). After washing with PBS with 0.1% Tween-20, a second blocking step was performed with 10% NGS for 10 minutes.

For the Collagen alpha-1 (XVIII) chain staining, after antigen retrieval skin sections were incubated with 1% BSA, 22.52 mg/ml glycine in PBS with 0,1% Tween-20 for 1 h to block unspecific antibody binding. Then, samples were incubated with primary antibodies in blocking solution at 4 °C overnight. The mouse anti-Collagen alpha-1(XVIII) chain antibody (DB144-N2, 1:150)[67] was a kind gift from Dr. Ritva Heljasvaara from the University of Oulu (Finland). Sections were also incubated with rabbit anti-Vimentin (Cell Signaling, D21H3, 1:100). After washing with PBS with 0.1% Tween-20 we directly proceeded to secondary antibody incubation.

For Tetraspanin 8, Periostin and the Collagen alpha-1(XVIII) chain stainings, sections were then incubated with corresponding Alexa Fluor-conjugated secondary antibodies (Life Technologies, cat. no. A11034, A32732, and A21103) for 2 h at room temperature. Nuclear counterstaining was performed with DAPI and slides were mounted with Vectashield Antifade Mounting Medium (Vector Laboratories, cat. no. H-1000).

Images were taken with a Leica TCS SP5 (Leica Microsystems) confocal microscope using a ×40 oil immersion lens and were further processed using the Fiji software[66].

**Statistics and reproducibility**. Statistical analyses of the scRNA-seq data ($n = 5$, see Supplementary Table 1) were carried out using the CellRanger and Seurat packages in R, and Wilcoxon Rank Sum tests were used to perform gene expression comparisons between cell clusters. Two-sided two-proportion z-tests were also used to compare fibroblast proportions in G1 between young and old samples.

As stated above, RNA-FISH assays were performed in three biological replicates per gene, and for quantification of *APCDD1*- and *CTHRC1*-positive cells we used a two-way ANOVA test with Dunnett correction. To compare CD248 positive cells we used an unpaired two-sided *t*-test. IHC experiments were performed in 1–6 replicates per gene.

**Reporting summary**. Further information on research design is available in the Nature Research Reporting Summary linked to this article.

## Data availability

scRNA-seq datasets are available from the Gene Expression Omnibus (GEO) database (accession number GSE130973). Any other data are available from the corresponding authors upon reasonable request.

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

## Acknowledgements

We are indebted to Dirk Tönges and Miguel Sánchez Eimil for their personal support of this study. We gratefully acknowledge the help of Julika Dick and Enrico Streit for dermatosurgery, and the clinical staff of the Department of Dermatology at Heidelberg University Hospital. We also thank Ritva Heljasvaara for the Collagen alpha-1(XVIII) chain antibody, Katharina Röck and Marc Winnefeld for helpful advice and stimulating discussions, and Felix Bormann for computational help. This work was supported by a research grant from the Helmholtz Zukunftsthema 'Aging and Metabolic Programming' (AMPro) to F.L.

## Author contributions

L.S.-B., G.R., A.S.L., M.R.-P., and F.L. analyzed the data. L.S.-B. performed scRNA-seq with support from J.-P.M. and K.R. S.S., and L.S.-B. performed RNA FISH and IHC assays. A.S.L. provided clinical samples. M.R.-P. and F.L. conceived the study. L.S.-B., M.R.-P., and F.L. wrote the paper with input from other authors. All authors read and approved the final manuscript.

## Competing interests

The authors declare no competing non-financial interests but the following competing financial interests: F.L. received consultation fees from Beiersdorf AG. The other authors have no competing financial interests.
