## [Peer Review File · Communications Biology]

Reviewers' comments:

Reviewer #1 (Remarks to the Author):

Summary

The authors investigate fibroblast heterogeneity in human skin using the 10X droplet-based single-cell RNA sequencing (scRNA-seq) method. They analyzed whole skin samples from the ilioinguinal regions (a sun-protected area) of five male patients; 2 patients were 'young' (aged 25 and 27) and 3 were 'old' (aged 53-70). t-distributed Stochastic Neighbor Embedding (t-SNE) was used to visualize sequenced data and to identify cell clusters. Based on data pooled from all five patients, the authors identified four clusters as fibroblasts (5,913 cells in total) which highly expressed archetypal fibroblast markers (LUM, DCN, VIM, PDGFRA and COL1A2). The authors used Gene ontology (GO) to make predictions about the functions of distinct clusters, focusing fibroblasts from the young male patients (1,795 cells). Based on the GO analysis, and by looking at which clusters expressed genes known to be highly expressed in papillary and dermal fibroblasts from previous literature, the authors conclude there to be 4 populations of fibroblasts; secretory-papillary, secretory-reticular, mesenchymal and pro-inflammatory fibroblasts. They use immunofluorescence staining and confocal imaging to show the spatial location of the pro-inflammatory and secretory-reticular fibroblasts within human skin.

Overall

This paper is well-written and addresses the important and relevant topic of fibroblast heterogeneity in human skin. The major weakness of the work presented, however, is the absence of any functional assays to support the bold claims made about the existence and function of the four distinct fibroblast subpopulations described. Although clusters realized by t-SNE plots provide useful predictions, these need to be validated. The authors attempt to infer function and location of distinct fibroblasts from gene expression, however, without in vitro and in vivo assays the work in its current state is only a description of single cell RNA sequencing data on human skin cells and does not propel us any further forward from where the literature currently stands. While the authors stain for two of the fibroblast populations in skin (pro-inflammatory and secretory-reticular fibroblasts), it is unclear why this was not done for the mesenchymal and secretory-papillary fibroblasts. Consequently, much of what the authors write are overstatements that have not been rigorously tested.

Abstract

- The abstract makes a number of bold statements which are not supported by the work presented. The following claims should be modified to more accurately represent the data in this paper:
 1. 'four main fibroblast subpopulations that can be spatially localized and functionally distinguished, as they exhibit specific secretory, mesenchymal and pro-inflammatory functions.'
 2. 'fibroblast 'priming' becomes reduced with age.'
 3. 'Our work thus provides comprehensive evidence for a functional specialization of human dermal fibroblasts and identifies the loss of cellular identity as a major age-related change'
- In addition, while the authors analyze the transcriptomes of >15,000 cells, the focus of their analysis was on fibroblasts, of which they analyzed 5,913 cells.

Introduction

- The authors stress the importance of investigating fibroblasts from sun-protected areas and but do not emphasize why this is important and nor do they describe what the phenotypic effects of aged skin are, please elaborate.
- The authors cite work on describing both human and mouse fibroblasts interchangeably. While some

knowledge can be extrapolated from mouse fibroblast subpopulations, there are numerous differences in skin physiology and wound healing between mice and humans which likely confer species-specific differences cellular subtypes. Can the authors separate knowledge gained from work conducted on mice and human fibroblasts in the introduction.

- While the authors claim to be looking at sun-protected areas, this has not been validated, and if the patients spent time without clothes on in the sun (E.g. as a kid or adult) or use sunbeds, this statement would be weakened. Was this information checked?
- Again, the following statement should be modified as it is not supported by the data: presented: 'Transcriptomic analysis of the young fibroblasts indicated the presence of four main populations that could be spatially localized and functionally distinguished'

Figure 1

- Do Merkel and Langerhans cells fall into the two Keratinocyte clusters (#6 and #7)? Where are the nerve cells, glandular cells (e.g. apocrine and sweat)?
- The authors mention 'The anatomy of the skin can vary considerably depending on a number of endogenous and environmental factors'– Do the authors find the same 14 populations in total and 4 populations of fibroblasts when each patient's data was analyzed separately? Are the same clusters found when comparing young versus old?

Figure 2

- The authors use genes known to be expressed in reticular or papillary fibroblasts to hypothesize whether the clusters are reticular or papillary. However, it should be emphasized these are hypotheses that have not been validated. The subsequent claim that 'Taken together, these results reveal 'priming' of human skin fibroblasts into four functionally and spatially defined populations' cannot be made until the identity and spatial location of these fibroblasts has been validated by immunostaining.
- The authors further make claims about the function of the 4 clusters of fibroblasts 'Two populations have prominent roles in the generation of structural collagen and ECM organization. One of these populations is clearly associated with the papillary dermis... and the other with the reticular dermis ... A third population, which is mainly distributed within the reticular dermis, appears to maintain a greater mesenchymal potential and contains the dermal papilla cells...our analysis also identifies a fourth population with pro-inflammatory functions and a mostly reticular localization'. Again, these claims regarding the identity, spatial location, and functions of these four fibroblast clusters identified by TSNE plots needs validation before such statements can be made.

Figure 3

- The authors aimed to identify representative markers for each subpopulation, however, although TSPAN8 is detectable on the cell surface, the ligand CCL19 is not. Identification of unique surface molecules is more useful as it enables prospective isolation of cell populations by flow cytometry.
- Furthermore, have the authors attempted to specifically interrogate the gene expression data for the markers in the Philippeos et al. and Tabib et al papers? Understanding how congruent this new data is with previous published reports is useful to help build the over picture of human fibroblast heterogeneity.
- It is unclear why the authors performed immunofluorescence staining of only CCL19 and TSPAN8 with vimentin in healthy human skin. While they are able to confirm the existence of pro-inflammatory cells and secretory-reticular population of fibroblasts, the identity and location of the two remaining populations of fibroblasts is undetermined.
- Can the authors provide quantification of the Immunofluorescence staining in the reticular and papillary dermis to support the images provided

Figure 4

- In the old versus young comparison could it not be the case that this 'age-dependent loss of

functional identity' is purely related to the fact that 3 samples from old patients of various ages were compared to 2 samples from younger patients. Furthermore, aged skin is subjected to more environmental exposures which can confound these findings. To validate whether or not this is the case, the authors should repeat by comparing each aged dataset to the pooled younger dataset.

- Again, the claim that 'gene expression signatures, as papillary fibroblasts became less papillary and more reticular' – is not necessarily represented by the data, and there is no immunofluorescent staining to support this finding.

Figure 5

- Can the authors define the abbreviation SAASP in text and figure legends.
- Based on the expression of cell-surface receptors and their interacting ligands, the authors claim 'most interactions identified in young fibroblasts are potentially lost during intrinsic skin aging.' – however, again, due to the likely variability between samples from aged individuals, a subsequent analysis comparing each patient's dataset to the overall mean from young people would strengthen this statement.
- The authors make claims of interactions between fibroblasts and communicating cell types based purely on expression of cell-surface receptors and their interacting ligands. The authors should look at the secretome and at the receptors and ligands in corresponding cell types to support their hypotheses.

Discussion

- The discussion is also populated with bold claims, unsupported by the data presented.
- The authors state 'We notice that we did not detect any evidence for adipogenic differentiation in our dataset' – however, no adipogenic assays were performed
- The authors state 'However, whole human skin has not been analyzed at a comprehensive level yet, as previous single-cell studies were either focused on sun-exposed material obtained from a heterogeneous group of donors²² or provided limited coverage from flow-sorted cells' and 'Our study analyzes single-cell transcriptomes from more than 15,000 skin cells that were all obtained from the same sun-protected location and only from healthy male donors.' – however, they focus on fibroblasts, so these statements (claiming to analyze whole skin) are misleading.
- Again, authors claim 'In addition, we also describe several fibroblast subpopulations with distinct functional and spatial specifics.' – the functional and spatial specifics have not been validated.
- Page eleven: after the colon, semicolons should be used to separate the points.
- The authors state 'Similarly, it is well known that fibroblasts can differentiate into other mesenchymal cell types^{48,49}, which is reflected by our mesenchymal population.' And 'We notice that we did not detect any evidence for adipogenic differentiation in our dataset' Yet the authors have not tested the differentiation potential of their 'mesenchymal population'.
- The authors state 'Importantly, our analysis of the interactome of each fibroblast subpopulation shows that aging causes a considerable decrease in all their potential interactions, and especially those that are maintained with undifferentiated keratinocytes' and then 'This may provide a plausible explanation for the thinning and flattening of the dermal-epidermal junction, which constitutes one of the most important phenotypic effects of aged skin¹⁷.' However, it is unclear how this assumption was made.
- The authors should mention the limitations of 10X genomics (e.g. less genes can be covered, although more cell transcriptomes can be profiled).

Method

- Skin specimens: please state the age of each sample skin was collected from and whether they had any comorbidities (e.g. systemic sclerosis or any condition affecting the skin). Are the skin samples mentioned in this section separate from those used for scRNA-seq – this is not clear, please address.
- Please define the abbreviation 'MACS' when used for the first time.

- Catalog numbers should be provided for all kits and reagents used.

Reviewer #2 (Remarks to the Author):

In this manuscript, Solé-Boldo et al use single-cell RNA-seq to explore the heterogeneity of full-thickness human skin from a sun-protected area of young and old healthy donors. They focus on dissecting the currently less-well defined fibroblast compartment, and aim to elucidate age-related changes within different fibroblast sub-populations. While this is a valuable dataset with important questions asked, major computational re-analyses and conformation in situ, as outlined below, are essential prior publication.

Major concerns:

1. Gene expression differences attributed to inter-individual differences

It is well known that inter-individual differences in humans cause cells to separate according to donor. This separation is often smaller than cell-type separations but large enough to obscure sub-type identification. The authors here analyzed two young and three old individuals without taking donor-related gene expression differences and clustering into consideration. This could unfortunately affect several of the core conclusions. To solve these concerns, the authors need to perform re-analyses of their data to assess to what extent the identification of the four fibroblast clusters is caused by donor differences. The specifics are detailed below.

- Figure 1. Please add a panel in Fig 1a where the cells are colored based on the donor. Even though those differences are not expected to be stronger than cell-type differences, it is important to explicitly demonstrate that in each new data set.

- Figure 2. The authors separate fibroblasts from two individuals into four putative sub-types (Fig 2a). To what extent are the identified four cluster differences driven by donor differences? It is very likely that sub-type clusters could be due to inter-individual differences that would confound many of the core conclusions of this manuscript. First, the authors need to explore to what extent the identification of the four clusters in Fig 2a is caused by donor differences. They need to adopt a computational strategy to normalize away donor differences, which is challenging but necessary. The authors also need to add details on the computational analysis of how clustering and low-dimensional projections were performed. For example, specify how tSNE was constructed. How many variable genes were used? Which clustering algorithm in Seurat was used? How were the fibroblasts sub-clustered, if at all? Sub-clustering of cell types most often results in more "pure" clusters, however it seems that the authors decided to base the four sub-clusters on their main clustering (Fig. 1) instead of sub-clustering fibroblasts separately, if so why?

Fig 2d: It does not seem that the four identified clusters agree very well with papillary and reticular fibroblasts, it rather looks like a "north-south division" with perhaps a stronger enrichment for cluster I and III, respectively. The authors write: "In contrast, fibroblasts in clusters II, III and IV showed more prominent expression of the reticular signature, with the highest reticular expression levels observed in cluster III (Fig. 2d)." but no statistical test was made to see whether this are significant differences. Again, this may relate to significant individual-based clustering of fibroblasts that obscures the "real underlying" sub-types of fibroblasts.

- Page 9: "Taken together, these results reveal 'priming' of human skin fibroblasts into four

functionally and spatially defined populations." Even after investigating whether there are really four sub-types within their data, the authors need to update the statements related to (1) spatial positioning of the four sub-types as they have not been shown in this manuscript; (2) what is the meaning of the word "priming" in this context; (3) the use of the word "functional" should be limited to when function has been assessed; i.e. it is not appropriate to use it for sub-types with some "functional categories" differing in Gene Set Enrichment analyses.

2. Age-related effect

The computational analysis strategy used (the "young" and "old" samples were initially normalized separately and later merged?) makes it fully plausible that the observed age-related effects could have technical nature. Importantly, the observed differences could be confounded by inter-individual differences existing in the two "young" individuals, which naturally the older unrelated donors fail to recapitulate (i.e. causing a worse fit for old cells to map to the four clusters identified in the "young" cells). Therefore, this part of the manuscript can not be assessed before the whole computational analysis is redone to (1) normalize old and young samples together and (2) clearly demonstrate which clusters are due to inter-individual differences. Also here, please add the information on the (1) number of most variable genes used for the analysis, (2) exact clustering method in Seurat used, (3) an assessment whether the identified numbers of clusters are robust (or whether it could really be a few more or less sub-types without any power to distinguish how many they actually were).

3. Spatial validation of fibroblast sub-clusters

Based on identified marker genes from their single-cell analysis (Fig. 2), the authors should show the spatial distribution/location of the defined fibroblast subpopulations in situ on both young and old skin tissue sections (via IHC and/or RNA-FISH staining). This is an important independent validation that tremendously strengthens the claims and thus the entire study.

Minor concerns:

Page 6: "scRNA-seq using the commercial version of the Drop-seq protocol²⁷." This is very unclear regarding the single-cell RNA-seq method used. It is better to directly specify in the main text that the 10X Genomics platform (v2 chemistry) was used.

How were doublets identified/removed, if at all? There seem to be cells having different identities within some clusters, e.g. there are T-cell populations with a mixed identity of erythrocytes, keratinocytes or fibroblasts.

In supplementary Fig 9, the authors subtract DP cells from the mesenchymal population. How was that done and why only for this figure? Moreover, there should be hair follicle cells in the dataset as there are dermal papilla cells, however these cells aren't mentioned anywhere and not included in the study?

Reviewer #1

1. The abstract makes a number of bold statements which are not supported by the word presented. The following claims should be modified to more accurately represent the data in this paper:

> The abstract was edited to tone down the 'bold statements'.

Introduction

2. The authors stress the importance of investigating fibroblasts from sun-protected areas and but do not emphasize why this is important and nor do they describe what the phenotypic effects of aged skin are, please elaborate.

> The importance of investigating fibroblasts from sun-protected areas has now been emphasized at the beginning of the Results section, as well as in the Introduction section in regard to the work of Tabib *et al.* Two new references on photoaging and its effects on skin (25, 26) have also been included in the revised version of the manuscript.

3. The authors cite work on describing both human and mouse fibroblasts interchangeably. While some knowledge can be extrapolated from mouse fibroblast subpopulations, there are numerous differences in skin physiology and wound healing between mice and humans which likely confer species-specific differences cellular subtypes. Can the authors separate knowledge gained from work conducted on mice and human fibroblasts in the introduction.

> The corresponding part of the introduction was edited to separate the mouse-related knowledge from the human-related knowledge.

4. While the authors claim to be looking at sun-protected areas, this has not been validated, and if the patients spent time without clothes on in the sun (E.g. as a kid or adult) or use sunbeds, this statement would be weakened. Was this information checked?

> It is indeed possible to have episodes of intermittent UV-exposure of the ilioinguinal region, for example, with indoor or outdoor nude tanning. Also, patients were not specifically queried about the use of sun-beds, practice of naturism/nudism or childhood sun exposure, which is challenging to quantify retrospectively. However, all patients in our study underwent a full body skin examination by a dermatologist prior to surgery and medical records were reviewed with a particular focus on skin diseases and skin-affecting co-morbidities. Also, none of the patients had a history of UV-therapy. Finally, none of the patients showed clinical evidence of acute or chronic actinic skin damage (e.g. elastosis, hyperkeratosis, pigmentation changes) or tanned skin in the ilioinguinal region at the time of surgery.

5. Again, the following statement should be modified as it is not supported by the data: presented: 'Transcriptomic analysis of the young fibroblasts indicated the presence of four main populations that could be spatially localized and functionally distinguished'

> This statement has been toned down in the introduction and at all other positions in the text. We now describe the fibroblast subpopulations to have distinct 'functional annotations' throughout the revised manuscript. In addition, we included RNA FISH and immunohistochemistry assays that spatially localize the four main fibroblast subpopulations in the dermis (Fig. 3 and Supplementary Figs. 5-6).

Figure 1

6. Do Merkel and Langerhans cells fall into the two Keratinocyte clusters (#6 and #7)? Where are the nerve cells, glandular cells (e.g. apocrine and sweat)?

> These cell types represent very minor subpopulations in our samples that may become separated only after higher-level clustering. For example, we can detect Schwann cells within the melanocyte cluster, Langerhans cells within one of the macrophages/dendritic cell cluster and smooth muscle cells (arrector pili) within one of the pericyte clusters. We now provide this information (plus additional plots with glandular epithelium cells and keratinocytes from the hair follicle) in Supplementary Fig. 1d. A complete analysis of all minor cell types would be beyond the scope of our study, which focuses on fibroblast subpopulations.

7. The authors mention 'The anatomy of the skin can vary considerably depending on a number of endogenous and environmental factors'– Do the authors find the same 14 populations in total and 4 populations of fibroblasts when each patient's data was analyzed separately? Are the same clusters found when comparing young versus old?

> We have repeated the data analysis for individual donors and included the results in Supplementary Fig. 2. Importantly, although the number of clusters vary slightly, we essentially find the same 14 populations (9 cell types) and 4 fibroblasts subpopulations in all individual skin samples. In addition, the revised version of the manuscript now also includes a figure showing the cell contribution of each donor to the clusters defined by the integrated analysis (Supplementary Fig. 1a). Also see Reviewer #2, point 1.

Figure 2

8. The authors use genes known to be expressed in reticular or papillary fibroblasts to hypothesize whether the clusters are reticular or papillary. However, it should be emphasized these are hypotheses that have not been validated. The subsequent claim that 'Taken together, these results reveal 'priming' of human skin fibroblasts into four functionally and spatially defined populations' cannot be made until the identity and spatial location of these fibroblasts has been validated by immunostaining.

> The spatial localization of the four subpopulations has now been extensively validated by RNA FISH and immunohistochemistry. The corresponding results are shown in Fig. 3 and Supplementary Figs. 5-6. Nevertheless, a proper functional analysis remains to be done and we have therefore toned down the statement to describe the subpopulations as having different 'functional annotations' (also see point 5 above). Also see Reviewer #2, point 4.

9. The authors further make claims about the function of the 4 clusters of fibroblasts 'Two populations have prominent roles in the generation of structural collagen and ECM organization. One of these populations is clearly associated with the papillary dermis... and the other with the reticular dermis ... A third population, which is mainly distributed within the reticular dermis, appears to maintain a greater mesenchymal potential and contains the dermal papilla cells...our analysis also identifies a fourth population with pro-inflammatory functions and a mostly reticular localization'. Again, these claims regarding the identity, spatial location, and functions of these four fibroblast clusters identified by TSNE plots needs validation before such statements can be made.

> See point 8 above.

Figure 3

10. The authors aimed to identify representative markers for each subpopulation, however, although TSPAN8 is detectable on the cell surface, the ligand CCL19 is not. Identification of unique surface molecules is more useful as it enables prospective isolation of cell populations by flow cytometry.

> We were not able to identify cell surface markers that were specific enough for each of our subpopulations. Table 1 shows the most representative markers of each fibroblast subpopulation, as well as the percentages of cells that express the gene in that particular group. Although CCL19 is a secreted cytokine, it can also be detected in the cytoplasm and thus served as a useful marker for the validation of our pro-inflammatory fibroblast population.

11. Furthermore, have the authors attempted to specifically interrogate the gene expression data for the markers in the Philippeos *et al.* and Tabib *et al.* papers? Understanding how congruent this new data is with previous published reports is useful to help build the overall picture of human fibroblast heterogeneity.

> We have now included the expression of the marker genes defining fibroblast subpopulations in Philippeos *et al.* and Tabib *et al.* in Supplementary Fig. 14. Some of the described genes were not detectable in our dataset, while others were not specific for one of our fibroblast subpopulations. This may suggest that the previously described subpopulations and their marker genes are specific for the skin regions that were analyzed previously. It is also possible that the higher analytical power and/or more defined sample selection of our study contributed to these differences.

12. It is unclear why the authors performed immunofluorescence staining of only CCL19 and TSPAN8 with vimentin in healthy human skin. While they are able to confirm the existence of pro-inflammatory cells and secretory-reticular population of fibroblasts, the identity and location of the two remaining populations of fibroblasts is undetermined.

> See point 8 above.

13. Can the authors provide quantification of the Immunofluorescence staining in the reticular and papillary dermis to support the images provided

> This quantification is now provided in Fig. 3a and Supplementary Fig. 5a for the new RNA FISH assays included in the revised version of the manuscript.

Figure 4

14. In the old versus young comparison could it not be the case that this 'age-dependent loss of functional identity' is purely related to the fact that 3 samples from old patients of various ages were compared to 2 samples from younger patients. Furthermore, aged skin is subjected to more environmental exposures which can confound these findings. To validate whether or not this is the case, the authors should repeat by comparing each aged dataset to the pooled younger dataset.

> We have performed this analysis (fibroblasts from the 2 young samples combined vs. fibroblasts from each individual old sample) and we essentially obtained the same results. Two illustrative examples (age-related decrease in global collagen generation and the changes in the spatial gene expression signatures) are now provided in Supplementary Fig. 8 of the revised manuscript.

15. Again, the claim that 'gene expression signatures, as papillary fibroblasts became less papillary and more reticular' –is not necessarily represented by the data, and there is no immunofluorescent staining to support this finding.

> Our claim is now supported by statistical analyses (pairwise comparisons using Wilcoxon rank sum tests), see Fig. 2d.

Figure 5

16. Can the authors define the abbreviation SAASP in text and figure legends.

> Done.

17. Based on the expression of cell-surface receptors and their interacting ligands, the authors claim ‘most interactions identified in young fibroblasts are potentially lost during intrinsic skin aging.’—however, again, due to the likely variability between samples from aged individuals, a subsequent analysis comparing each patient’s dataset to the overall mean from young people would strengthen this statement.

> We have now analyzed the interactions in each sample independently and the results are shown in Supplementary Figure 12. The results show that the two young samples are highly similar and clearly distinct from the two 70 y.o. samples. The 54 y.o. sample showed an intermediate phenotype.

18. The authors make claims of interactions between fibroblasts and communicating cell types based purely on expression of cell-surface receptors and their interacting ligands. The authors should look at the secretome and at the receptors and ligands in corresponding cell types to support their hypotheses.

> This will be an important point for future analyses, but we consider it outside the scope of our present study.

Discussion

19. The discussion is also populated with bold claims, unsupported by the data presented.

> All statements that are not supported by the data were toned down.

20. The authors state ‘We notice that we did not detect any evidence for adipogenic differentiation in our dataset’ – however, no adipogenic assays were performed

> Supplementary Fig. 15 now shows that we do not detect any expression of the adipogenesis genes *PPARG*, *PPARD* and *FABP4*, which were previously found overexpressed in mouse fibroblasts upon aging (Salzer *et al.*, 2019).

21. The authors state ‘However, whole human skin has not been analyzed at a comprehensive level yet, as previous single-cell studies were either focused on sun-exposed material obtained from a heterogeneous group of donors²² or provided limited coverage from flow-sorted cells’ and ‘Our study analyzes single-cell transcriptomes from more than 15,000 skin cells that were all obtained from the same sun-protected location and only from healthy male donors.’ – however, they focus on fibroblasts, so these statements (claiming to analyze whole skin) are misleading.

> This has been clarified in the text, also see point 1 above.

22. Again, authors claim ‘In addition, we also describe several fibroblast subpopulations with distinct functional and spatial specifics.’—the functional and spatial specifics have not been validated.

> See point 8 above.

23. Page eleven: after the colon, semicolons should be used to separate the points.

> Corrected.

24. The authors state 'Similarly, it is well known that fibroblasts can differentiate into other mesenchymal cell types^{48,49}, which is reflected by our mesenchymal population.' And 'We notice that we did not detect any evidence for adipogenic differentiation in our dataset' yet the authors have not tested the differentiation potential of their 'mesenchymal population'.

> The statements were corrected and toned down. We now write: 'Similarly, it is well known that fibroblasts can differentiate into other mesenchymal cell types^{53, 54}, which is reflected by the predicted attributes of our mesenchymal population' and 'we did not detect an upregulation of adipogenesis genes'.

25. The authors state 'Importantly, our analysis of the interactome of each fibroblast subpopulation shows that aging causes a considerable decrease in all their potential interactions, and especially those that are maintained with undifferentiated keratinocytes' and then 'This may provide a plausible explanation for the thinning and flattening of the dermal-epidermal junction, which constitutes one of the most important phenotypic effects of aged skin¹⁷.' However, it is unclear how this assumption was made.

> The statement was toned down and simplified. We removed the assumption and now write "This may represent a previously unknown molecular feature of skin aging".

26. The authors should mention the limitations of 10X genomics (e.g. less genes can be covered, although more cell transcriptomes can be profiled).

> This is now described in the first paragraph of the Results section: "This commercial version of the high-throughput Drop-seq protocol²⁷ is capable of identifying cell populations by analyzing the expression of highly expressed genes in a high number of cells."

Methods

27. Skin specimens: please state the age of each sample skin was collected from and whether they had any comorbidities (e.g. systemic sclerosis or any condition affecting the skin). Are the skin samples mentioned in this section separate from those used for scRNA-seq – this is not clear, please address.

> The age of each skin sample subjected to single-cell RNA sequencing is specified in both the Results section and Supplementary Table 1. The age of the specimens used for the validation assays is stated in the figure legends (Fig. 3 and Supplementary Figs. 5-6) and in the Methods section (RNA FISH and immunohistochemistry). Clinical aspects are now described in the Methods section (Clinical samples). For single-cell RNA sequencing samples as well as for the old samples used for validation assays, we now state "All patients underwent a full body skin examination by a dermatologist prior to surgery and medical records were reviewed with a particular focus on skin diseases and skin-affecting comorbidities. No clinical evidence or a history of an inflammatory or systemic skin disease (e.g. systemic sclerosis, lupus erythematosus), or co-morbidities typically affecting the skin, or UV-sensitivity of the skin (e.g. chronic immunosuppression, chronic renal failure) was recorded". For the young samples used for validation assays, we now state: 'These samples were taken from non-sun-exposed body areas of healthy (male and female) individuals with no present co-morbidities'.

28. Please define the abbreviation 'MACS' when used for the first time.

> The name MACS is a registered trademark of Miltenyi Biotech, not an abbreviation.

29. Catalog numbers should be provided for all kits and reagents used.

> Catalog numbers have been added in the Methods section. As it seems unusual to supply the numbers for all kits and reagents, this was done for key/biological reagents.

Reviewer #2

Gene expression differences attributed to inter-individual differences

It is well known that inter-individual differences in humans cause cells to separate according to donor. This separation is often smaller than cell-type separations but large enough to obscure sub-type identification. The authors here analyzed two young and three old individuals without taking donor-related gene expression differences and clustering into consideration. This could unfortunately affect several of the core conclusions. To solve these concerns, the authors need to perform re-analyses of their data to assess to what extent the identification of the four fibroblast clusters is caused by donor differences. The specifics are detailed below.

1. Figure 1. Please add a panel in Fig 1a where the cells are colored based on the donor. Even though those differences are not expected to be stronger than cell-type differences, it is important to explicitly demonstrate that in each new data set.

> This result has now been included (Supplementary Fig. 1a). Also see Reviewer #1, point 7.

2. Figure 2. The authors separate fibroblasts from two individuals into four putative sub-types (Fig 2a). To what extent are the identified four cluster differences driven by donor differences? It is very likely that sub-type clusters could be due to inter-individual differences that would confound many of the core conclusions of this manuscript. First, the authors need to explore to what extent the identification of the four clusters in Fig 2a is caused by donor differences. They need to adopt a computational strategy to normalize away donor differences, which is challenging but necessary. The authors also need to add details on the computational analysis of how clustering and low-dimensional projections were performed. For example, specify how tSNE was constructed. How many variable genes were used? Which clustering algorithm in Seurat was used? How were the fibroblasts sub-clustered, if at all? Sub-clustering of cell types most often results in more “pure” clusters, however it seems that the authors decided to base the four sub-clusters on their main clustering (Fig. 1) instead of sub-clustering fibroblasts separately, if so why?

> As shown in our new Supplementary Fig. 1A, all the 4 fibroblast subpopulations are present in all the 5 samples, and therefore unlikely to be driven by donor differences. In addition, we have now included additional details on the integration, normalization and clustering in the Methods section. This also includes all the requested details such as the number of variable genes (2,000 genes) and algorithm (shared nearest neighbor modularity optimization based clustering) used for clustering the cells.

Subclustering of the fibroblasts was indeed achieved at the first level and identified the 4 well-defined subpopulations that are described in our manuscript. Further sub-clustering is possible at higher levels, but usually derived from the splitting of the original subpopulations. For example, the second-level clustering of the fibroblasts from the young samples resulted in 5 fibroblast subpopulations, since the pro-inflammatory population was divided into two distinct pro-inflammatory clusters. Similarly, the individual analyses sometimes also separated the dermal papilla stem cells, a well-known mesenchymal cell population defined by the expression of markers such *CRABP1* and *TNN*, from the rest of the mesenchymal fibroblasts (Supplementary Figs. 4b-c).

3. Fig 2d: It does not seem that the four identified clusters agree very well with papillary and reticular fibroblasts, it rather looks like a “north-south division” with perhaps a stronger enrichment for cluster I and III, respectively. The authors write: “In contrast, fibroblasts in clusters II, III and IV showed more prominent expression of the reticular signature, with the highest reticular expression levels observed in cluster III (Fig. 2d).” but no statistical test was made to see whether this are significant differences. Again, this may relate to significant individual-based clustering of fibroblasts that obscures the “real underlying” sub-types of fibroblasts.

> For establishing the dermal location of our fibroblast groups we used genes that were previously described to be differentially expressed between papillary and reticular fibroblasts in several publications (see Supplementary Fig. 3). A statistical test (pairwise comparisons using Wilcoxon rank sum test) has now been added to Fig. 2d and substantiates our previous conclusions.

4. Page 9: “Taken together, these results reveal ‘priming’ of human skin fibroblasts into four functionally and spatially defined populations.” Even after investigating whether there are really four sub-types within their data, the authors need to update the statements related to (1) spatial positioning of the four sub-types as they have not been shown in this manuscript; (2) what is the meaning of the word “priming” in this context; (3) the use of the word “functional” should be limited to when function has been assessed; i.e. it is not appropriate to use it for sub-types with some “functional categories” differing in Gene Set Enrichment analyses.

> The four subpopulations have now been extensively validated in terms of spatial localization by RNA FISH and immunohistochemistry. The corresponding results are shown in Fig. 3 and Supplementary Figs. 5-6. Nevertheless, a proper functional analysis remains to be done and we have therefore toned down the statement to describe the subpopulations as having different ‘functional annotations’. Also see Reviewer #1, points 5 and 8.

‘Priming’ is now defined in the sentence that follows the description of the 4 subpopulations: “Taken together, these results suggest that the functional annotation differences between subpopulations may reflect their ‘priming’ for distinct functional roles”.

Age-related effect

5. The computational analysis strategy used (the “young” and “old” samples were initially normalized separately and later merged?) makes it fully plausible that the observed age-related effects could have technical nature. Importantly, the observed differences could be confounded by inter-individual differences existing in the two “young” individuals, which naturally the older unrelated donors fail to recapitulate (i.e. causing a worse fit for old cells to map to the four clusters identified in the “young” cells). Therefore, this part of the manuscript cannot be assessed before the whole computational analysis is redone to (1) normalize old and young samples together and (2) clearly demonstrate which clusters are due to inter-individual differences. Also here, please add the information on the (1) number of most variable genes used for the analysis, (2) exact clustering method in Seurat used, (3) an assessment whether the identified numbers of clusters are robust (or whether it could really be a few more or less sub-types without any power to distinguish how many they actually were).

> Also see point 2 above. As shown in our new Supplementary Fig. 1a, all the 4 fibroblast subpopulations are present in all the 5 samples, and are therefore unlikely to be confounded by inter-individual differences. In addition, we have included the requested technical details

and explanations on the computational analysis in the Methods section. Also see point 2 above.

Spatial validation of fibroblast sub-clusters

6. Based on identified marker genes from their single-cell analysis (Fig. 2), the authors should show the spatial distribution/location of the defined fibroblast subpopulations in situ on both young and old skin tissue sections (via IHC and/or RNA-FISH). This is an important independent validation that tremendously strengthens the claims and thus the entire study.

> The four subpopulations and their spatial distribution have now been extensively validated by RNA FISH and immunohistochemistry. The corresponding results are shown in Fig. 3 and Supplementary Figs. 5-6.

Minor concerns:

7. Page 6: “scRNA-seq using the commercial version of the Drop-seq protocol²⁷. “ This is very unclear regarding the single-cell RNA-seq method used. It is better to directly specify in the main text that the 10X Genomics platform (v2 chemistry) was used.

> This has now been specified in the first paragraph of the Results section.

8. How were doublets identified/removed, if at all? There seem to be cells having different identities within some clusters, e.g. there are T-cell populations with a mixed identity of erythrocytes, keratinocytes or fibroblasts.

> An explanation on all quality controls including how doublets were removed has now been added to the Methods section.

To establish the identity of each cell cluster, we have used the average expression of 3 to 5 literature-based marker genes. However, some of these genes are not exclusively expressed in a single cell type (for example, *VIM* is highly expressed by fibroblasts as well as by other cell types such as macrophages, T-cells, pericytes or melanocytes), and this can increase the average expression in additional clusters. However, we believe that the selected literature-based markers clearly show and define the distinct cell types identified in this study.

9. In supplementary Fig 9, the authors subtract DP cells from the mesenchymal population. How was that done and why only for this figure? Moreover, there should be hair follicle cells in the dataset as there are dermal papilla cells, however these cells aren't mentioned anywhere and not included in the study?

> See our explanation for point 2 above and our new Supplementary Fig. 4. Also see Reviewer #1, point 6. Many additional minor subpopulations are present and could be identified by higher-level clustering. However, a complete analysis of all these minor cell types would be beyond the scope of our study, which focuses on fibroblast subpopulations.

Reviewers' comments:

Reviewer #1 (Remarks to the Author):

We thank the reviewers for appropriately addressing many of the points we highlighted in the previous review. There are three remaining points:

Response to point 3: the authors still have not clarified in their introduction which of the reported papers refer to work investigating mouse versus human dermal fibroblasts. E.g. in the paragraph starting 'Besides their fundamental role' references 22 and 23 are cited without clarifying whether this work was done on mouse or human skin.

Response to point 4: Can the authors state in the methods section that (in addition to the full body skin examination by a dermatologist prior to surgery and review of medical records with a particular focus on skin diseases and skin-affecting co-morbidities) that no patients had a history of UV-therapy showed clinical evidence of acute or chronic actinic skin damage (e.g. elastosis, hyperkeratosis, pigmentation changes) or tanned skin in the ilioinguinal region at the time of surgery. These details are not currently included.

Response to point 10 (related to figure 3): Can the authors explain in the body of the text that they were not able to identify cell surface markers that were specific enough for each of our subpopulations. This information is useful for other authors aiming to identify the human fibroblast subpopulations.

Reviewer #2 (Remarks to the Author):

The authors have made an effort to address the raised concerns, which has significantly improved the manuscript. I have a few remaining comments/suggestions mainly concerning "young vs old" fibroblasts:

1) Please provide a supplementary table with genes that are differentially expressed between old and young (side by side; including gene expression levels and a significance test for each gene).

2) Text line 230: "In agreement with the reduced proliferative capacity of aged cells, the expression profiles from old fibroblasts indicated a delay at the G1/S transition of the cell cycle in all four subpopulations (Fig. 4b).

This statement is not supported by Fig. 4b. There is almost no difference between Y and O (no statistical test was done) and for the population "MES" this is clearly not the case.

3) The most apparent difference between YOUNG and OLD fibroblast populations is a decrease of #9 (Fig. 2a vs Fig. 4a). It would be of interest to mention this observation in the manuscript if this is significant.

4) As the difference of YOUNG vs OLD fibroblasts is an important point/claim of the manuscript, it is important to include, or to refer to, stainings that indeed show a difference in gene expression in situ in respective fibroblast population(s).

Minor suggestions:

Fig. 3a/Fig. S5a: Papillary/Reticular region detail. As it is labeled now one thinks it's a detail from the above images. Could you add a word e.g. "additional" such as "Papillary (additional detail)"?

Fig. 3c: Could you please add in the figure the text "DP" next to the circle. As this is the DP region of the HF, the label on the left side is supposedly "HF bulb" instead of "HF bulge"?

- Please re-label "Subject 1, 2, 3 etc. to Donor 1, 2, 3" in all figures incl. supplement.

- To my knowledge, no one has yet claimed a definite "DP stem cell" population. If so, it would be better to call the subpopulation a "potential DP progenitor" or just "DP-associated cell population".

Reviewer #1 (Remarks to the Author):

We thank the [authors] for appropriately addressing many of the points we highlighted in the previous review. There are three remaining points:

Response to point 3: the authors still have not clarified in their introduction which of the reported papers refer to work investigating mouse versus human dermal fibroblasts. E.g. in the paragraph starting 'Besides their fundamental role' references 22 and 23 are cited without clarifying whether this work was done on mouse or human skin.

We have mentioned the organisms used in each reference. All works cited in the introduction include the requested information.

Response to point 4: Can the authors state in the methods section that (in addition to the full body skin examination by a dermatologist prior to surgery and review of medical records with a particular focus on skin diseases and skin-affecting co-morbidities) that no patients had a history of UV-therapy showed clinical evidence of acute or chronic actinic skin damage (e.g. elastosis, hyperkeratosis, pigmentation changes) or tanned skin in the ilioinguinal region at the time of surgery. These details are not currently included.

We can confirm that no patient had a history of UV-therapy, showed clinical evidence of acute or chronic actinic skin damage or presented tanned skin in the inguinoiliac region at the time of surgery. This sentence is now included in the methods section.

Response to point 10 (related to figure 3): Can the authors explain in the body of the text that they were not able to identify cell surface markers that were specific enough for each of our subpopulations. This information is useful for other authors aiming to identify the human fibroblast subpopulations.

This information has been now included in the section describing the RNA FISH and immunohistochemistry results.

Reviewer #2 (Remarks to the Author):

The authors have made an effort to address the raised concerns, which has significantly improved the manuscript. I have a few remaining comments/suggestions mainly concerning "young vs old" fibroblasts:

1) Please provide a supplementary table with genes that are differentially expressed between old and young (side by side; including gene expression levels and a significance test for each gene).

The requested table is now included as Suppl. Table 6.

2) Text line 230: "In agreement with the reduced proliferative capacity of aged cells, the expression profiles from old fibroblasts indicated a delay at the G1/S transition of the cell cycle in all four subpopulations (Fig. 4b). This statement is not supported by Fig. 4b. There is almost no difference between Y and O (no statistical test was done) and for the population "MES" this is clearly not the case.

Statistical tests have now been included in Fig. 4b. The results showed that there is a significant age-related G1/S delay in the secretory-papillary and pro-inflammatory subpopulations, while the observed differences for the secretory-reticular and mesenchymal subpopulations were not significant. The text, figure and legend have been updated accordingly.

3) The most apparent difference between YOUNG and OLD fibroblast populations is a decrease of #9 (Fig. 2a vs Fig. 4a). It would be of interest to mention this observation in the manuscript if this is significant.

We were unable to experimentally confirm the apparent reduction in the mesenchymal subpopulation (Fig. 4a), as this requires a high number of properly oriented hair follicles in tissue sections. This is now explained in the text.

4) As the difference of YOUNG vs OLD fibroblasts is an important point/claim of the manuscript, it is important to include, or to refer to, stainings that indeed show a difference in gene expression in situ in respective fibroblast population(s).

As an example, we have now included RNA FISH results that show the downregulation of the SAASP-related gene *CD248* in old secretory-reticular fibroblasts. See Supplementary Fig. 11c.

Minor suggestions:

Fig. 3a/ Fig. S5a: Papillary/Reticular region detail. As it is labeled now one thinks it's a detail from the above images. Could you add a word e.g. "additional" such as "Papillary (additional detail)"?

They are indeed a detail from the images above. We have clarified this in the figure legends.

Fig. 3c: Could you please add in the figure the text "DP" next to the circle. As this is the DP region of the HF, the label on the left side is supposedly "HF bulge" instead of "HF bulge"?

We have now added the "DP" next to the circle, as well as changed "HF bulge" for "HF bulb" here and in Supplementary Fig. 5c.

-Please re-label "Subject 1, 2, 3 etc. to Donor 1, 2, 3" in all figures incl. supplement.

Done.

-To my knowledge, no one has yet claimed a definite "DP stem cell" population. If so, it would be better to call the subpopulation a "potential DP progenitor" or just "DP-associated cell population".

Following the reviewer's suggestion, we have now changed "dermal papilla stem cells" for "dermal papilla-associated fibroblasts" throughout the manuscript.